# Acute repeated cage exchange stress modifies urinary stress and plasma metabolic profiles in male mice

Sayuri Fushuku[1], Miharu Ushikai[1], Emi Arimura[1,2], Yuga Komaki[1], Masahisa Horiuchi[1]*

1 Department of Hygiene and Health Promotion Medicine, Graduate School of Medical and Dental Sciences, Kagoshima University, Kagoshima, Japan, 2 Major in Food and Nutrition, Department of Life and Environmental Science, Kagoshima Prefectural College, Kagoshima, Japan

☯ These authors contributed equally to this work.
* masakun@m.kufm.kagoshima-u.ac.jp

## Abstract

Exposure to a novel environment is psychologically and physically stressful for humans and animals. The response has been reported to involve enhanced sympathetic nervous system activity, but changes in nutrient levels under stress are not fully understood. As a form of exposure to a novel environment, repeated cage exchange (CE, four times at 2-h intervals for 8 h from 08:00 h) during the light phase with no restraint on movement was applied to A/J mice, a strain particularly prone to stress. Body temperature was measured with a temperature-sensing microchip implanted in the interscapular region. The stress conditions and anxiety level were evaluated by measuring urinary catecholamines and corticosterone and by performing an anxiety-like behavior test, respectively. Major nutrients such as glucose, fatty acids, and amino acids in the plasma were also examined. CE mice showed a significant increase in body temperature with each CE. They also showed a significantly greater reduction of body weight change, more water intake, and higher levels of urinary catecholamines and corticosterone and anxiety-like behavior score than control mice. The model revealed a significantly lower plasma glucose level and higher levels of several essential amino acids, such as branched-chain amino acids and phenylalanine, than those of control mice. Meanwhile, free fatty acids and several amino acids such as arginine, aspartic acid, proline, threonine, and tryptophan in both sets of mice were significantly decreased from the corresponding levels at 08:00 h, while similar plasma levels were exhibited between mice with and without CE. In conclusion, repeated CE stress was associated with changes in glucose and amino acids in plasma. Although further study is needed to clarify how these changes are specifically linked to anxiety-like behavior, this study suggests the potential for nutritional intervention to counter stress in humans exposed to novel environments.

**Data Availability Statement:** All relevant data are within the paper and its Supporting Information files.

**Funding:** This work was supported by grants from the Ministry of Education, Culture, Sports, Science, and Technology of Japan (#20K10430 to Miharu Ushikai) and funded by the Kodama Memorial Fund for Medical Science Research. This work was also supported by the Core Research Program for "Preventive Medicine Core Unit" by Kagoshima University (to Masahisa Horiuchi). The funders had no role in study design, data collection and analysis, decision to publish, or preparation of the manuscript.

**Competing interests:** The authors have declared that no competing interests exist.

## Introduction

Modern life involves stresses such as difficulties in human relationships, pressure to perform at work, and exposure to new environments [1–3]. Such stresses may be related to the onset and development of mental health disorders, which are thought to occur in a manner dependent on the intensity of the stressor and the resilience of the subject [4, 5]. Alongside efforts to reduce stressors themselves, people's resilience to stress should also be considered. Limited information on the relationship between resilience to stress and nutritional factors, including amino acids, has been reported, even though amino acid-related compounds and minerals are thought to confer stress resilience [6, 7]. The occurrence of changes in nutrient levels upon exposure to stress would suggest the importance of dietary/lifestyle interventions for those facing stress. Therefore, the present study examined nutritional factors, including amino acids, in a novel stressed context.

Several procedures for applying psychological and physical stress to rodents, involving exposure to an altered environment, have been reported [8, 9]. These procedures include isolating mice or housing two mice together to generate dominant and subordinate mice. Regarding energy expenditure, dominant mice have a significantly greater energy expenditure than subordinate mice at the end of chronic psychosocial stress procedures. Additionally, subordinate mice show higher plasma glucose and non-esterified fatty acid levels than dominant mice after stress procedures. However, information on amino acid profiles is limited. In addition, in studies using an immobilization model where mice were restrained in a tube for several hours, the levels of most amino acids in plasma were reported to be decreased [10], and several amino acids were altered depending on the age of mice and duration of stress [11]. Moving mice to another cage can also be stressful. This has been reported to result in an increase in body temperature (BT) for several hours [12]. The importance of increased BT in response to stress may be that it alters brain function for fight-or-flight [13]. In the present study, movement from one cage to another was applied to establish a new environmental stress model. The response to this stress was examined by monitoring the autonomic nervous system and endocrine system. As an acute response to stress, enhanced activity of the autonomic nervous system and endocrine system leads to increases in blood and urinary catecholamines [12]. These increases indicate the enhanced catabolism of nutrients under stress conditions. Indeed, changes in nutrients upon exposure to stress have been reported for glucose and lipid metabolism, but the effects on amino acid metabolism have remained unclear. If a link exists between psychological stress and nutrients, it may be possible to use nutrients to manage psychologically stressful conditions. To achieve this goal, the changes in nutrients that occur under psychological stress should be determined. Moreover, psychological conditions are potentially linked to the prognosis of metabolic diseases such as diabetes. Relief of psychological conditions by nutritional intervention might be important in managing metabolic diseases.

In this paper, we report the changes in nutrients, especially amino acids, which have limited information regarding their responses to stress and are consumed to meet energy demand. We used a stress model in which the animals can still move freely, namely, repeated cage exchange (CE), mimicking exposure to a novel environment.

## Materials and methods

### Experimental design

Six-week-old male A/J mice were purchased from Japan SLC, Inc. (Shizuoka, Japan). The mice were housed individually in a humidity- and temperature-controlled (50 ± 10%, 22 ± 2˚C) facility under a 12-h light/dark cycle (07:00–19:00 h). The mice were housed with animal

bedding (Japan SLC, Inc.) and provided with *ad libitum* access to water and food (CE2; CLEA Japan, Inc., Tokyo, Japan) before undergoing the experiments. Mice that weighed between 25 and 32 g at the start of the experiment were used. To measure BT, a temperature-sensing microchip (Digital Angel Corporation, St. Paul, MN, USA) was implanted into the interscapular region of mice anesthetized with ketamine (75 mg/kg) and medetomidine (1 mg/kg) by i.p. injection [14, 15]. The animals were allowed to recover for 1 week before any experimental manipulation. The BT was measured noninvasively by a specialized apparatus (POCKET READER; Destron Fearing, TX, USA) [14]. For the collection of pure urine without contamination by feces, awake mice were held with the left hand. They then naturally or with a short compression of the abdomen by the right hand urinated on a clean wrap [16]. The voided urine was collected, and the specimen was stored at −80˚C for later analysis. At the end of the experimental procedure, the mice were anesthetized using pentobarbital (100 mg/kg), and blood was collected from the heart. The blood was mixed with EDTA (final concentration of 4 mM) and centrifuged (3000 g for 10 min at room temperature), and the supernatant was stored at −80˚C for later analysis. Organs, including the heart and liver, white adipose tissue (WAT) as fat surrounding the epididymis, and brown adipose tissue (BAT) as fat near the neck were weighed. To determine the reference values, mice were measured at 08:00 h under fed conditions. A total of 72 mice were used to obtain the results for Fig 1 (10 control (CT) and 10 CE mice), Fig 2 (10 different CT and 10 different CE mice), Tables 1 and 3 (six different mice were used to obtain data at 08:00 h, six CT and six CE mice from the experiment for Fig 2 were used in 3 weeks after the experiment), and Table 2 (eight different CT and eight different CE mice). For the water intake experiment, 5 CT and 5 CE mice were used (S1 Fig). This study was approved by the Ethics Committee for Animal Experimentation at Kagoshima University (approval numbers MD20122 and MD21066), in accordance with the Japanese national guidelines for animal experiments.

## Application of physical stress by repeated CE

The stress protocol was only performed once a day and is an acute stress paradigm. Mice were transferred to a different cage four times at intervals of 2 h. At the first transfer at 08:00 h, a new cage without any bedding was used. For the second to fourth transfers, the mice were transferred to a cage where other mice had lived. At the first transfer at 08:00 h, access to food was removed, after which the mice were fasted for 8 h. To ensure consistent food and water conditions across all groups, CT mice were fasted for 8 h concomitantly with the stressed mice. The body weight was measured before and after fasting with and without CE. The amount of water consumed was measured by weighing the water apparatus (Hydropac®; Natsume Seisakusho Co., Ltd., Tokyo, Japan) on a digital scale before and after the experiment (16:00 h) and calculating the difference [17]. In the measurement of water intake, based on the previously published method [17], the cage cover for CT mice was removed every 2 h, as was that for CE mice.

## Behavioral testing

The elevated plus maze test was performed to evaluate anxiety-like behavior. The open and closed arms represent unsafe and safe environments, respectively. Spending more time in the open arms indicates mice are more resistant to anxiety-like behavior [18]. Behavioral testing of the mice using the elevated plus maze was performed at 16:00 h after the procedure with or without CE. This maze consisted of a black Plexiglas cross (arms: 25 cm long × 5 cm wide) elevated 50 cm above the floor. Two opposite arms were enclosed by black walls (25 cm long × 15 cm high), while the two other arms were open. Each mouse was placed in the center of the

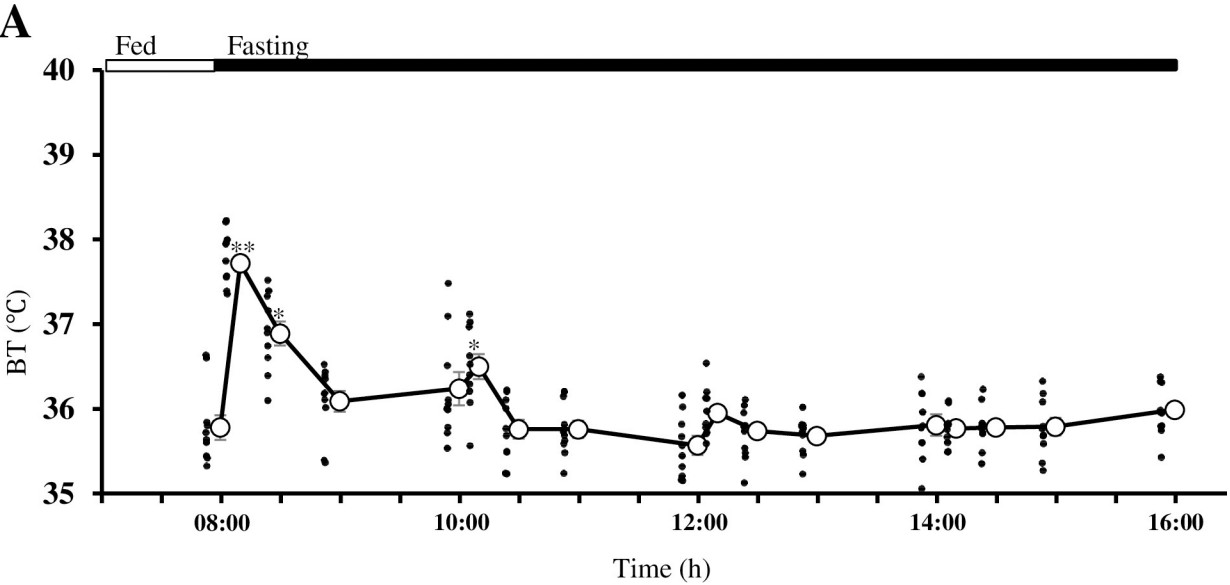

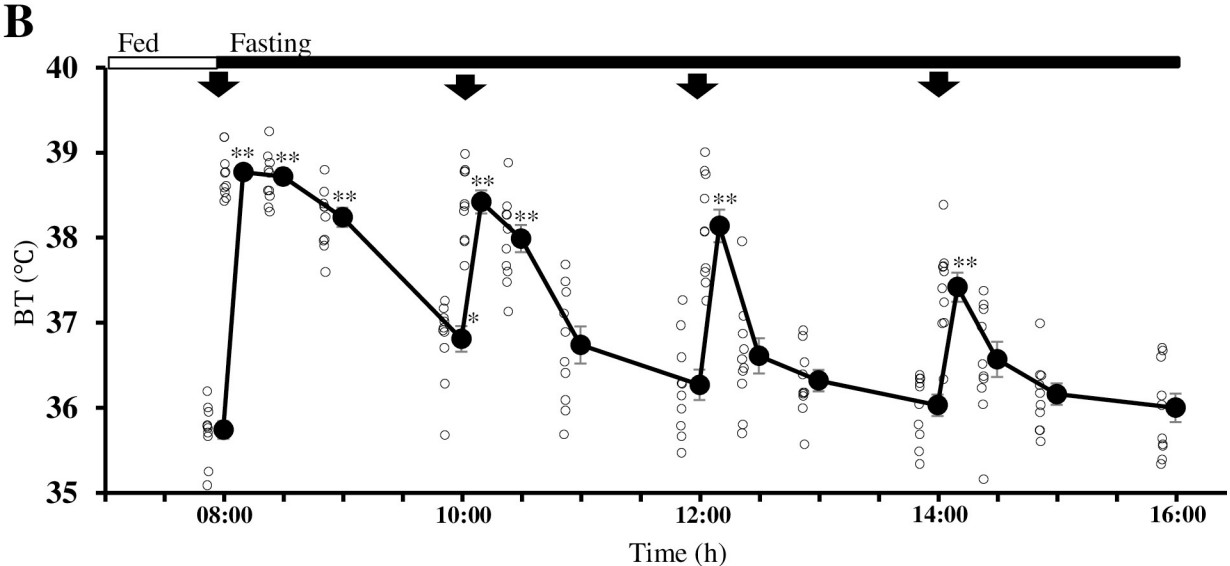

**Fig 1.** Change in body temperature in mice without (A) or with (B) repeated cage exchange. The mean values for mice without (CT, large open circle) and with cage exchange (CE, large closed circle) are presented. Arrows in B denote the time when cage exchange was performed. Both sets of mice were under fed and fasting (8 h) conditions as indicated by the long open and closed boxes, respectively. The data are presented as the mean ± SEM (n = 10). The raw data (CT, small closed circles; CE, small open circles) are presented as dots in the figure. The data were analyzed statistically by one-way repeated measures ANOVA followed by Bonferroni's multiple comparisons test as a *post hoc* test. Differences are shown as $^{**}p < 0.01$ compared with the value of the respective mice at 08:00 h. Two-way repeated measures ANOVA showed a significant difference in the interaction between CT mice and CE mice during the time course ($F_{(16, 288)}$ = 22.6, $p < 0.01$).

apparatus facing a closed arm and observed for 5 min [18]. The numbers of entries into the open and closed arms and the time spent in the open arms were recorded. The total number of entries into the arms (closed plus open), the ratio of open to total arm entries, and the time in the closed arms were calculated. The procedure was performed under a light condition of 15 lx

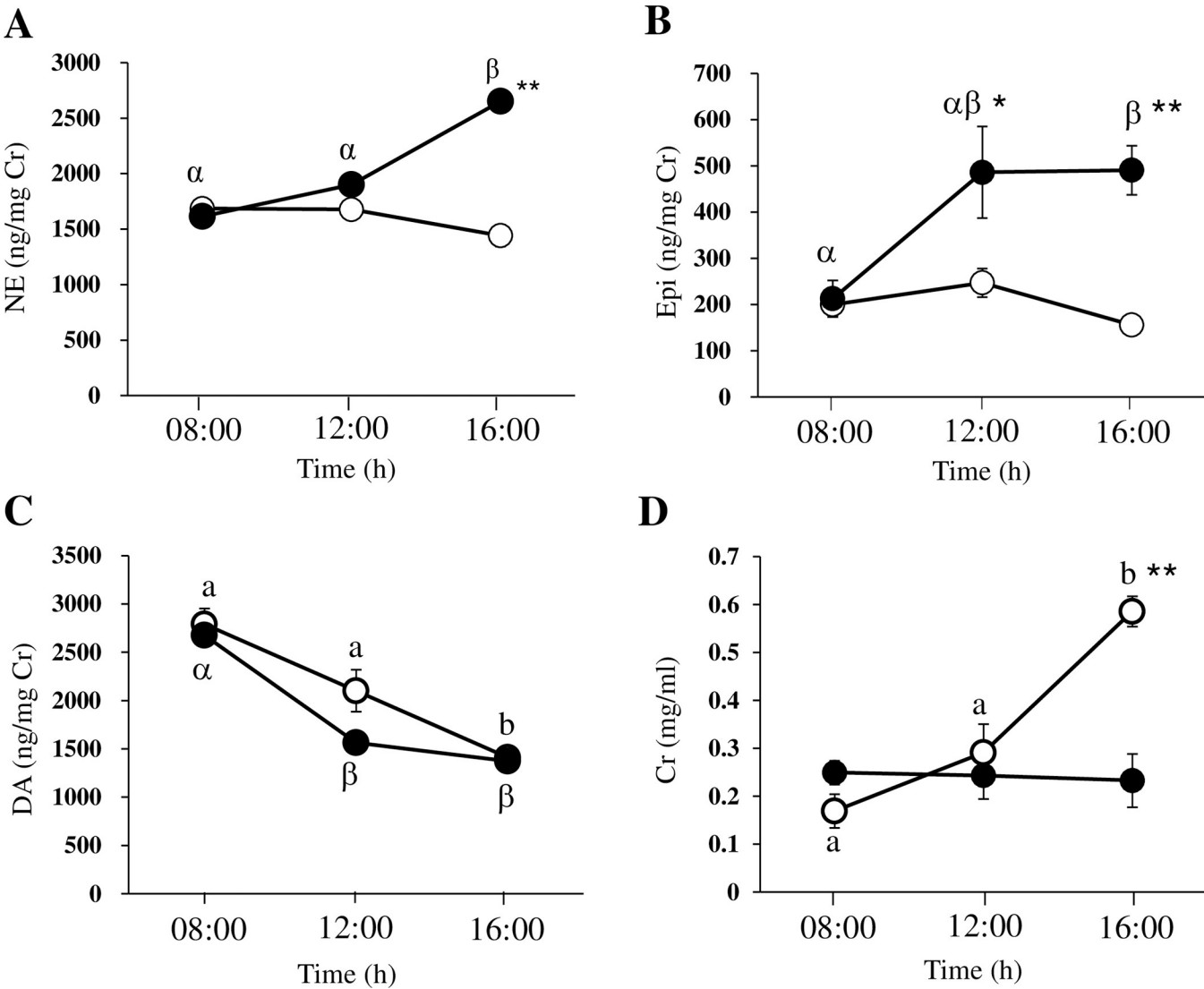

**Fig 2. Changes in urinary catecholamines and creatinine during the cage exchange procedure.** For norepinephrine (A, NE), epinephrine (B, Epi), and dopamine (C, DA), the value is expressed as ng of substance/mg of creatinine (Cr). Urinary Cr is presented in (D). The concentration or ratio means for mice without (CT, open circle) and with cage exchange (CE, closed circle) are shown. The data are presented as the mean ± SEM (n = 7–10 samples). The data were analyzed statistically by the Mann–Whitney $U$ test or Kruskal–Wallis test, followed by the Steel–Dwass test as a *post hoc* test. Significant differences are shown as $*p < 0.05$, $**p < 0.01$, compared with CT mice at the respective timepoints. Different symbols (English letters for CT, Greek letters for CE) represent significant differences ($p < 0.05$) among the values of CT or CE mice at the timepoints.

measured by a digital illuminometer (LX-100, Satotech, Kanagawa, Japan). The study was recorded on digital media using a specialized camera (Logicool C615n HD webcam, Logicool, Tokyo, Japan). Then, the recorded data were analyzed using ANYmaze software (Stoelting, Wood Dale, IL, USA).

## Biochemical measurements

Plasma glucose and free fatty acids (FFAs) were measured in 2 μl and 4 μl of plasma, respectively, based on an enzymatic procedure, using commercial kits (LabAssay Glucose and LabAssay NEFA; FUJIFILM Wako Pure Chemical Corp., Osaka, Japan), in accordance with the manufacturer's instructions. Insulin and glucagon were measured in 5 μl and 10 μl of

**Table 1. Body weight changes in CT and CE mice.**

| | | CT mice | | | CE mice | | |
|---|---|---|---|---|---|---|---|
| **Body weight** | | | | | | | |
| 08:00 h | g | 28.70 | ± | 0.62 | 28.55 | ± | 0.77 |
| 16:00 h | g | 27.25 | ± | 0.63## | 26.29 | ± | 0.66## |
| Change | g | −1.45 | ± | 0.07 | −2.26 | ± | 0.11** |

Data are expressed as the mean ± SEM ($n$ = 6 per group).

Differences are shown as **$p < 0.01$ compared with CT mice (unpaired $t$-test) and ##$p < 0.01$ compared with the respective mice at 08:00 h (paired $t$-test). The comparison between CT and CE mice over the time course was performed using two-way repeated measures ANOVA ($F_{(1, 10)}$ = 38.17, $p < 0.01$).

CT, control; CE, cage exchange.

plasma, respectively, using ELISA kits (Morinaga Institute of Biological Science Inc., Kanagawa, Japan, and FUJIFILM Wako Pure Chemical Corp.). Creatinine (Cr) was measured in 2 μl of urine using a kit based on the creatininase reaction (CRE-IV; KAINOS Lab. Inc., Tokyo, Japan). Amino acid concentrations were measured using an amino acid analyzer and the ninhydrin reaction (JLC-500/V; JEOL Ltd., Tokyo, Japan). For the amino acid measurement, 80 μl of plasma deproteinized with 5% (w/v) sulfosalicylic acid was used [19]. Urine (20 μl) was used for the measurement of catecholamines by the following procedure. Using a MonoSpin PBA column (GL Sciences, Tokyo, Japan), urine supplemented with an internal standard (isoproterenol) was purified in accordance with the manufacturer's instructions [20]. The purified urine (corrected by the recovery rate using the internal standard) was subjected to HPLC for the measurement of norepinephrine (NE), epinephrine (Epi), and dopamine (DA) with reference to a standard [21]. The values were corrected based on urinary Cr for different concentrations of the urine specimens [22].

## Statistical analysis

Data are expressed as the mean ± standard error of the mean (SEM). For differences between two groups, Student's $t$-test or Welch's $t$-test was performed for data with equal or unequal homogeneity of variance, respectively. A paired $t$-test was performed for comparisons of values within groups. A Mann–Whitney $U$ test was performed for non-normally distributed data under the unpaired condition. Statistical analysis of the changes in BT was performed using one-way repeated measures analysis of variance (ANOVA). Then, Bonferroni's multiple comparisons test was performed as a *post hoc* comparison for the data in Fig 1. Two-way repeated measures ANOVA was performed for the data in Fig 1 and Table 1, as appropriate. The Kruskal–Wallis test was performed for non-normally distributed data under the unpaired condition because of a lack of urine sample collection for the data in Fig 2. Then, the Steel–Dwass test was performed as a *post hoc* comparison. All statistical analyses were performed using EZR (ver.1.61), and differences were considered significant at $p < 0.05$ [23]. We omitted some data and mice according to outlier tests, including the Smirnov–Grubbs test and QQ plots.

## Results

### Physiological parameters such as BT, body weight, and water intake

To establish this mouse stress model, we performed four CEs at 2-h intervals after 08:00 h. BT was measured at 0, 10, 30, 60, and 120 min after each CE. As shown in Fig 1 and S1 Table, the mice showed a significant increase in BT associated with each CE. The value increased rapidly over 10 min, after which it gradually decreased over 2 h to almost the baseline level. In the case

**Table 2. Psychological parameters in CT and CE mice at 16:00 h after the respective procedures.**

| | | CT mice | | | CE mice | | |
|---|---|---|---|---|---|---|---|
| **Elevated plus maze test** | | | | | | | |
| Open arm entries | times | 6.5 | ± | 1.1 | 1.4 | ± | 1.0** |
| Closed arm entries | times | 27.1 | ± | 4.6 | 20.6 | ± | 1.9 |
| Total entries | times | 33.6 | ± | 3.9 | 22.0 | ± | 2.4* |
| Open arm entries/total entries | % | 22.0 | ± | 5.1 | 4.9 | ± | 3.0** |
| Open arm duration | s | 7.5 | ± | 1.8 | 3.0 | ± | 2.2* |

Data are expressed as the mean ± SEM (n = 8 per group).

Differences are shown as *$p < 0.05$, **$p < 0.01$ compared with CT mice (Mann–Whitney $U$ test).

CT, control; CE, cage exchange.

of the BT without CE, the values were significantly higher at 08:10 h than at 08:00 h, possibly because of food removal. Regarding BW, as shown in Table 1 and S3 Table, both CE and CT mice showed significant decreases compared with their values at baseline (08:00 h). CE mice showed lower BWs than CT mice, but this did not reach significance. The change in BW of the respective mice with CE was significantly lower than the value of CT mice. CE mice showed significantly greater water intake than CT mice (n = 5, 0.120 ± 0.017 g vs. 0.022 ± 0.007 g, $p < 0.05$). In addition, CE mice showed a significantly higher corticosterone level in urine collected at the end of the experimental period than that in CT mice (n = 9, 3.36 ± 0.53 μg/mg Cr vs. 1.13 ± 0.19 μg/mg Cr, $p < 0.05$).

## Urinary catecholamines

CE mice showed significantly higher NE and Epi values at 16:00 h and at 12:00 h and 16:00 h, respectively, compared with those at 08:00 h than CT mice (Fig 2 and S2 Table). Compared with those at 08:00 h, CE mice showed significantly higher NE and Epi values at 16:00 h. Meanwhile, both sets of mice showed similar DA levels at particular timepoints. In addition, DA significantly decreased during the experimental period for both sets of mice. The two sets of mice showed different changes in Cr, suggesting that the CEs significantly influenced the urinary Cr concentration, which is related to urine condensation.

## Behavior as evaluated by the elevated plus maze test

CE mice showed significantly fewer entries into the open arms and total arms than CT mice, as shown in Table 2 and S4 Table. In terms of the percentage of entries into the open arms and the time spent in the open arms, CE mice showed significantly lower values than CT mice.

## Organ/tissue weights

The weights of organs/tissues such as the liver, heart, WAT, and BAT were measured at 08:00 h as reference values and at 16:00 h. These variables did not differ significantly between CE and CT mice at 16:00 h (Table 3, S3 and S5 Tables). Compared with the values at 08:00 h, both groups of mice showed significantly lower liver and BAT weights, and the ratios of liver /BW and BAT/BW, at 16:00 h.

## Nutrients in plasma

Parameters reflecting nutrients and their related hormones such as carbohydrates, lipids, and amino acids in plasma were measured (Table 3). CE mice showed significantly lower plasma

**Table 3. Organ/tissue weights and nutrient-related biochemical parameters in plasma.**

| | | | | | | | CT mice | | | CE mice | |
|---|---|---|---|---|---|---|---|---|---|---|---|
| Time | | | 08:00 h | | | | 16:00 h | | | 16:00 h | |
| **BW** | g | 28.27 | ± | 0.77 | | 27.25 | ± | 0.63 | 26.29 | ± | 0.66 |
| **Organ weights** | | | | | | | | | | | |
| Liver | g | 1.618 | ± | 0.058 | | 1.151 | ± | 0.028## | 1.087 | ± | 0.065## |
| Heart | g | 0.110 | ± | 0.003 | | 0.106 | ± | 0.004 | 0.105 | ± | 0.003 |
| WAT | g | 0.576 | ± | 0.077 | | 0.548 | ± | 0.068 | 0.532 | ± | 0.064 |
| BAT | g | 0.138 | ± | 0.012 | | 0.096 | ± | 0.007# | 0.081 | ± | 0.010## |
| Liver/BW | | 0.057 | ± | 0.001 | | 0.042 | ± | 0.001## | 0.041 | ± | 0.002## |
| Heart/BW | | 0.004 | ± | 0.000 | | 0.004 | ± | 0.000 | 0.004 | ± | 0.000 |
| WAT/BW | | 0.020 | ± | 0.002 | | 0.020 | ± | 0.002 | 0.020 | ± | 0.002 |
| BAT/BW | | 0.005 | ± | 0.000 | | 0.004 | ± | 0.000## | 0.003 | ± | 0.000# |
| **Biochemical parameters** | | | | | | | | | | | |
| Glucose | mg/dl | 231 | ± | 12 | | 224 | ± | 14 | 174 | ± | 12*,## |
| FFAs | μEq/l | 503 | ± | 43 | | 307 | ± | 25## | 320 | ± | 34## |
| Glucagon | pmol/l | 4.50 | ± | 0.66 | | 3.34 | ± | 0.25 | 3.41 | ± | 0.44 |
| Insulin | ng/ml | 1.23 | ± | 0.24 | | 0.72 | ± | 0.10 | 0.69 | ± | 0.11 |
| HOMA-IR | | | | | | 18.8 | ± | 3.5 | 15.5 | ± | 3.4 |
| G/I ratio | ng/pmol | 0.11 | ± | 0.03 | | 0.11 | ± | 0.02 | 0.12 | ± | 0.02 |
| **Amino acids** | | | | | | | | | | | |
| Alanine | nmol/ml | 281.1 | ± | 16.3 | | 274.1 | ± | 25.8 | 255.1 | ± | 30.4 |
| Arginine | nmol/ml | 114.2 | ± | 8.1 | | 73.4 | ± | 3.9## | 67.3 | ± | 3.9## |
| Asparagine | nmol/ml | 25.4 | ± | 1.3 | | 28.1 | ± | 2.2 | 26.9 | ± | 1.5 |
| Aspartic acid | nmol/ml | 11.4 | ± | 0.8 | | 8.1 | ± | 0.6## | 7.2 | ± | 0.4## |
| Cysteine | nmol/ml | 55.1 | ± | 2.7 | | 58.2 | ± | 4.6 | 58.2 | ± | 1.5 |
| Glutamic acid | nmol/ml | 28.5 | ± | 2.5 | | 22.9 | ± | 2.4 | 23.4 | ± | 1.4 |
| Glutamine | nmol/ml | 477.3 | ± | 29.2 | | 402.8 | ± | 14.4# | 434.7 | ± | 24.2 |
| Glycine | nmol/ml | 189.5 | ± | 3.1 | | 234.0 | ± | 8.8## | 208.0 | ± | 11.4 |
| Histidine | nmol/ml | 66.6 | ± | 3.4 | | 67.4 | ± | 7.0 | 60.6 | ± | 3.6 |
| Isoleucine | nmol/ml | 105.3 | ± | 7.4 | | 69.0 | ± | 6.4## | 96.1 | ± | 6.7* |
| Leucine | nmol/ml | 153.3 | ± | 12.3 | | 106.3 | ± | 10.2# | 153.8 | ± | 10.3** |
| Lysine | nmol/ml | 235.3 | ± | 14.4 | | 184.4 | ± | 17.7# | 214.4 | ± | 10.7 |
| Methionine | nmol/ml | 44.1 | ± | 2.3 | | 40.3 | ± | 2.9 | 37.9 | ± | 2.1 |
| Phenylalanine | nmol/ml | 69.9 | ± | 3.1 | | 60.0 | ± | 3.3# | 75.8 | ± | 4.4* |
| Proline | nmol/ml | 71.2 | ± | 4.7 | | 53.5 | ± | 4.1# | 49.6 | ± | 2.6## |
| Serine | nmol/ml | 90.4 | ± | 3.9 | | 107.7 | ± | 8.7 | 92.0 | ± | 3.8 |
| Threonine | nmol/ml | 129.2 | ± | 6.7 | | 102.1 | ± | 6.5# | 97.6 | ± | 4.0## |
| Tryptophan | nmol/ml | 72.4 | ± | 2.8 | | 56.0 | ± | 3.8## | 62.0 | ± | 2.0# |
| Tyrosine | nmol/ml | 88.5 | ± | 5.0 | | 82.7 | ± | 6.1 | 74.1 | ± | 6.4 |
| Valine | nmol/ml | 269.0 | ± | 15.6 | | 173.6 | ± | 10.8## | 216.2 | ± | 13.0*,# |
| BCAAs | nmol/ml | 527.6 | ± | 35.1 | | 348.9 | ± | 26.9## | 466.1 | ± | 29.6* |

Data are expressed as the mean ± SEM ($n$ = 5–6 per group).

Differences are shown as *$p < 0.05$, **$p < 0.01$ compared with CT mice; #$p < 0.05$, ##$p < 0.01$ compared with the value at 08:00 h using an unpaired $t$-test, Welch's $t$-test, or the Mann-Whitney $U$ test, as appropriate (see supporting information).

BW, body weight; WAT, white adipose tissue; BAT, brown adipose tissue; FFAs, free fatty acids; HOMA-IR, homeostasis model assessment-insulin resistance; G/I ratio, glucagon/insulin ratio; BCAAs, branched-chain amino acids (isoleucine, leucine, valine); CT, control; CE, cage exchange. Among the amino acids, essential amino acids are underlined.

glucose levels than CT mice. However, there were no significant differences in plasma insulin, glucagon, the homeostasis model assessment-insulin resistance (HOMA-IR), and the glucagon/insulin (G/I) ratio between CE and CT mice. Compared with the value at 08:00 h, both sets of mice showed a lower plasma insulin value, while CE mice also showed a significantly lower plasma glucose value, at 16:00 h. Additionally, there was no significant difference in FFAs between CE and CT mice, although compared with the value at 08:00 h, both sets of mice showed significantly lower FFA values at 16:00 h. Meanwhile, CE mice showed significantly higher levels of isoleucine (Ile), leucine (Leu), phenylalanine (Phe), and valine (Val) than CT mice. Therefore, the level of branched chain amino acids (BCAAs) (sum of Ile, Leu, and Val) was significantly higher in CE mice than in CT mice. Compared with the value at 08:00 h, both sets of mice showed significantly lower values of arginine (Arg), aspartic acid (Asp), proline (Pro), threonine (Thr), tryptophan (Trp), and Val at 16:00 h.

## Discussion

The present study revealed that repeated CE leads to an increase in BT and a decrease in BW, along with an increase in urinary catecholamines, suggesting enhanced activity of the sympathetic nervous system. Repeated CE was also shown to be stressful, as revealed by the psychological test results, and associated with greater water intake and higher urinary corticosterone levels. Notably, higher concentrations of several amino acids, such as BCAAs and Phe, along with lower plasma glucose levels were detected under repeated CE. The significance of the higher concentrations of these amino acids regarding their relationships with the psychological state should be discussed. However, these measures (water drinking, BT, changes in amino acids) could also be attributed to increased physical activity. At present, we cannot differentiate the effects of stress from those of physical activity. The direct relationship between the changes in amino acids and psychological findings, including urinary stress and anxiety-like behavior test results, should be evaluated by interference experiments.

Among the stressors in everyday life and work, exposure to new environments has been thought of as a psychologically and physically stressful condition. A new environment has been reported to lead to increases in blood pressure and heart rate, in association with increased urinary catecholamines, in humans [24]. Psychological stress also induces an increase in BT in animals and humans [25]. The present study attempted to establish a mouse model of stress induced by repeated exposure to a new environment.

Given that the A/J strain has been reported to be the mouse strain most sensitive to physical and psychological stress, it was selected for the present study [26, 27]. In previous studies, a restraint procedure was applied to establish models of psychological stress [28, 29]. However, restraint stress suppresses drinking and eating, alongside the restriction of free movement. Therefore, we attempted to establish psychological and physical stress while maintaining freedom of the mice to drink and move. Single CE has been reported to lead to an increase in BT for approximately 2 h in mice [12]. In the present study, repeated CE led to an increase in BT at every CE (Fig 1), which to the best of our knowledge, is a novel finding. Additionally, repeated CE was associated with higher catecholamine excretion in urine and a higher anxiety-like behavior score (Fig 2 and Table 2).

Our findings suggest that CE induced activation of the sympathetic nervous system. NE and Epi are secreted from endings of sympathetic nerves and the adrenal gland, respectively. Stimulation of the sympathetic nervous system by stress might evoke stimulation of the adrenal gland, leading to increased secretion of Epi into the blood and urine [30]. Indeed, the repeated CE stress applied here was associated with increased excretion of NE and Epi into the urine even at 12:00 h (after two CEs). The changes in NE and Epi were maintained at 16:00 h

(after four CEs), suggesting that CE performed four times is sufficient to evoke stress under conditions in which the sympathetic nervous system is stimulated. Enhancement of sympathetic nervous system activity may increase BT through BAT [13]. Interestingly, the Cr concentration in urine at 16:00 h in CE mice was lower than that in CT mice. The repeated CE stress may suppress vasopressin, which was elevated in the fasting condition and/or according to the circadian rhythm in CT mice. In human studies, patients with psychiatric symptoms showed polyuria, which may be related to the inappropriate secretion of antidiuretic hormones such as vasopressin [31]. The mice exposed to repeated CE showed a worse score in the elevated plus maze test (Table 2). Additionally, CE mice had a significantly higher urinary corticosterone level than CT mice (see Results). Taken together, the findings obtained in this study suggest that repeated CE is an appropriate approach to establish an animal model to examine stress and its effects on the body.

Animal stress models subjected to restraint or blood collection showed increased blood glucose levels [32, 33]. Although the blood glucose level is physiologically maintained by hepatic glycogenolysis and gluconeogenesis, those systems are enhanced under stress conditions [32, 33]. However, the present study showed inconsistent results, revealing relatively low blood glucose levels during 8 h of fasting (Table 3). In particular, gluconeogenesis might not be enhanced in the present stress model. Interleukin (IL)-6 plays an important role in enhanced gluconeogenesis in model animals with stress induced by restraint or blood collection, and further studies will be required to examine IL-6 in the repeated CE stress model.

The present model is characterized by a lack of remarkable differences in the weights of organs/tissues (Table 3) upon comparing mice with and without CE. No significant difference in insulin was observed, but insulin levels were lower at 16:00 h than at 08:00 h in both groups of mice. At present, we speculate that the decrease in glycogen is the main cause of the lower weights of the liver and BAT at 16:00 h than at 08:00 h. The autonomic nervous system tone in response to low insulin levels has been reported to be associated with glycogen content in the liver and BAT, which might lead to decreases of the weights of the liver and BAT under fasting conditions [34]. The results in the present study are in line with this finding.

Regarding amino acids in plasma, the mice that underwent repeated CE showed higher levels of four amino acids (Ile, Leu, Val, and Phe) than the mice subjected only to fasting. In the literature, mice placed in a tube under restraint have been reported to show decreased levels of almost all amino acids except Trp [10], while rats subjected to foot shock also showed decreased levels of almost all amino acids [35]. The difference in findings compared with those in the present study could be explained by the different experimental conditions, especially that the stress applied in the previous studies might be associated with restriction of movement and with pain. In contrast, the repeated CE stress applied here involved conditions in which the mice had the freedom to move and drink *ad libitum*, and thus the degree of stress might have been more moderate than that under restraint stress.

The outcomes, such as increased BT induced by repeated CE, might be related to increased physical activity. However, in the literature, exercise inducing physical activity reduced plasma BCAA levels in diabetic mice [19] and normalized plasma BCAAs in healthy overweight humans [36]. At present, we suggest that the increased BCAA concentration in CE mice is not explained by increased physical activity. Because the four amino acids that exhibited changes in the present study are essential amino acids that are not biochemically synthesized in the body, their low levels are explained by the decrease in their usage in the body under fasting conditions (Table 3). BCAAs such as Ile, Leu, and Val are metabolized by skeletal muscle and BAT [37]. Given that exposure to the stress of a novel environment in this study was associated with an increase in BT, heat production may have been enhanced in BAT. To meet energy demands, enhanced usage of BCAAs might occur. However, a higher level of BCAAs was

detected in the present study (Table 3). As such, the stress conditions may have suppressed the degradation of BCAAs. One possible explanation for this is that enhanced transport of glucose through the activated sympathetic nervous system even under fasting conditions may suppress the transport of BCAAs into the BAT in CE mice [23]. To determine whether this is true and, if so, to understand the mechanism involved in this process, further study is required.

BCAAs and Phe are used as energy sources [38]. Among the four amino acids that were increased in CE mice, Val, Ile, and Phe are classified as gluconeogenic amino acids that are involved in maintaining the blood glucose level through gluconeogenesis under fasting conditions. The disturbance in catabolism of the amino acids might be related to the low level of plasma glucose, although this warrants further study. Compared with the reference values (at 08:00 h), both sets of mice showed significantly lower values of Arg, Asp, Pro, Thr, Trp, and Val at 16:00 h. These amino acids might be used even under stress conditions. The influence of those amino acids that showed changes in levels under stress conditions on indicators of stress, including the anxiety-like behavior score and urinary catecholamine level, will be evaluated in future work as well.

Moreover, the pathological significance of the higher level of BCAAs should be discussed. The higher level of BCAAs may contribute to inhibit the transport of aromatic amino acids such as Trp, Tyr, and Phe through the large neutral amino acid transporter in the blood–brain barrier [10, 39]. As a result, under the stress condition, inhibition of the entry of these aromatic amino acids into the brain may have altered the concentration of neurotransmitters there. While the level of Trp was unchanged in the stress group in this study, serotonin synthesized from Trp may have been decreased by the increase in BCAAs (Table 3). Serotonin is an important neurotransmitter that is related to anxiety and depression. A higher score in the anxiety-like behavior test is consistent with a possible lower level of serotonin following a higher level of BCAAs in plasma. With regard to Phe, which is also an essential amino acid, its higher level may also have been associated with its lower usage in the body. However, why this amino acid, along with BCAAs, exhibited a higher level remains unclear.

This study had several limitations. First, only male mice were used in this study. Generally, female mice are more susceptible to psychological and social stress [40, 41]. To clarify whether there is a sex difference in the response to the stress applied in this study, female mice should be examined in future work. Second, the A/J strain was used here because this strain has been reported to be susceptible to psychological and social stress [26, 27]. However, the response to the stress of repeated CE may differ among mouse strains. To understand the genetic factors responsible for the response to this stress, other strains should be examined in future work. Third, we performed only one behavioral test, the elevated plus maze. To evaluate anxiety or anxiety-like behavior precisely, another test should be added in the future. Finally, the model established in the present study involves acute stress. Therefore, a chronic stress model should also be examined. For example, a chronic study in which mice are exposed to CE four times a day for a much longer period, such as a week or a month, should be performed to provide findings that can be extrapolated better to human experiences.

## Conclusions

Repeated CE in mice was shown to produce enhanced activation of the sympathetic nervous system and increased anxiety-like behavior, in addition to a lower level of plasma glucose and higher levels of several amino acids, such as BCAAs and Phe. Although the relationship between heightened anxiety-like behavior and the changes in these nutritional factors should be analyzed, this study suggests the potential for nutritional intervention to counter stress in humans exposed to novel environments.

## Supporting information

**S1 Fig. Diagram of the way to use mice for the experiments.**
(TIF)

**S1 Table. Raw data of Fig 1.**
(XLSX)

**S2 Table. Raw data of Fig 2.**
(XLSX)

**S3 Table. Raw data of Tables 1 and 3.**
(XLSX)

**S4 Table. Raw data of Table 2.**
(XLSX)

**S5 Table. Parameters of the statistical analysis for Table 3.**
(XLSX)

## Acknowledgments

We thank Edanz (https://jp.edanz.com/ac) for editing a draft of this manuscript.

## Author Contributions

**Conceptualization:** Sayuri Fushuku, Miharu Ushikai, Masahisa Horiuchi.

**Data curation:** Sayuri Fushuku, Miharu Ushikai.

**Formal analysis:** Sayuri Fushuku, Miharu Ushikai.

**Funding acquisition:** Miharu Ushikai, Masahisa Horiuchi.

**Investigation:** Sayuri Fushuku, Miharu Ushikai.

**Methodology:** Sayuri Fushuku, Miharu Ushikai, Emi Arimura, Masahisa Horiuchi.

**Project administration:** Masahisa Horiuchi.

**Supervision:** Masahisa Horiuchi.

**Validation:** Sayuri Fushuku, Miharu Ushikai, Yuga Komaki.

**Visualization:** Sayuri Fushuku, Miharu Ushikai.

**Writing – original draft:** Sayuri Fushuku.

**Writing – review & editing:** Miharu Ushikai, Yuga Komaki, Masahisa Horiuchi.

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
