## [Decision Letter · Decision Letter 0]

30 Jun 2023

PONE-D-23-13098Biological responses including changes of amino acids in plasma in mice subjected to repeated cage exchange stressPLOS ONE

Dear Dr. Horiuchi,

Thank you for submitting your manuscript to PLOS ONE. After careful consideration, we feel that it has merit but does not fully meet PLOS ONE’s publication criteria as it currently stands. Therefore, we invite you to submit a revised version of the manuscript that addresses the points raised during the review process.

We look forward to receiving your revised manuscript.

Kind regards,

Yukiori Goto, Ph.D.

Academic Editor

PLOS ONE

Journal Requirements:

"This work was supported by grants from the Ministry of Education, Culture, Sports, Science, and Technology of Japan (#20K10430 to Miharu Ushikai) and funded by the Kodama Memorial Fund for Medical Science Research. This work was also supported by the Core Research Program for “Preventive Medicine Core Unit” by Kagoshima University (to Masahisa Horiuchi)."

Reviewers' comments:

Reviewer's Responses to Questions

**Comments to the Author**

1. Is the manuscript technically sound, and do the data support the conclusions?

Reviewer #1: Yes

Reviewer #2: No

Reviewer #3: Yes

2. Has the statistical analysis been performed appropriately and rigorously? 

Reviewer #1: Yes

Reviewer #2: No

Reviewer #3: I Don't Know

3. Have the authors made all data underlying the findings in their manuscript fully available?

Reviewer #1: No

Reviewer #2: No

Reviewer #3: Yes

4. Is the manuscript presented in an intelligible fashion and written in standard English?

Reviewer #1: Yes

Reviewer #2: No

Reviewer #3: No

5. Review Comments to the Author

Reviewer #1: This study suggests that repeated cage exchange induce physiological and biochemical alterations. Although the results sound very interesting, there are several concerns below that would help to fully appreciate the study.

Major concerns:

1. Water intakes could be one of crucial indicators reflecting physiological alterations induced by repeated cage exchange (CE). This issue has been described on line 186-187, but without data, even though it is also mentioned that water intakes were measured every 2 hours on line 131-133.

2. NE/DA and Epi/DA ratios were evaluated based on the reason that cage exchange influenced on Cr concentration (line 199-203), although other factors such as corticosterone was still evaluated relative to Cr. Since cage exchange influenced on Cr, it should affect all factors uniformly, such that it does not much make sense that NE and Epi are evaluated relative to DA but not others.

3. This study has reported that decreased liver and brown adipose tissues in both CE and CT models only in 8 hours, which is surprisingly fast metabolism beyond the ordinary understanding. Moreover, in the CE model, blood glucose level, but not glucagon and insulin, was decreased. Therefore, decreased liver and brown adipose tissues could not be explain only by glycogen contents. Providing details to explain this observation would be helpful to understand the study. Taking a ratio of tissue to whole body weights would also yield an additional insight on this issue.

Reviewer #2: The following study used repeated cage exchange in mice as a model of psychological stress to investigate the outcomes of stress exposure on body temp., organ weight, anxiety behaviors, and amino acid concentrations in urine. The manuscript is written clearly enough to follow and the experimental design seems appropriate, although there are some concerns about the statistical methods that require attention. My biggest concern with this study is the use of the cage exchange model of stress, as it contains many confounds that cannot be easily controlled for. This manuscript could be made suitable for publication, but would require an overhaul of the discussion and a very careful consideration of the rationale for the experimental treatments.

1) My major concern with this study is that the authors attribute their dependent outcomes to “psychological stress” due to repeated cage changes. This interpretation is problematic. Mice typically become more active in new environments, such as following cage changes. Can the authors tease apart the effects of stress from those of physical activity? Nearly every single dependent measure (e.g. water drinking, body temperature, changes to amino acids, cort levels, etc.) could just as easily be attributed to increased physical activity. This is an issue that I’m not sure can be readily addressed. Minimally, the authors may be required to restructure their arguments in the discussion to reflect this possible confound or provide evidence from literature why this might not be an issue for particular dependent measures (like with amino acid levels, if even possible). Second, the use of “psychological” to describe behavioral testing and outcomes, as well as “psychological stress” is further hard to tease apart using a rodent model. How do the authors know their outcomes are purely psychological?

2) Statistical considerations: 1) Test statistics are nearly completely absent from the results section. The authors should report their test statistics and make it clear which tests were being performed since the authors chose to perform parametric and non-parametric tests. 2) How did the authors determine the normality and non-normality of data? Most, if not all, of the measurements are continuous variables that should fall within a normal distribution. Thus, it is unclear how the assumption of normality is violated. 3) If I am following the statistical analysis section correctly, it seems that a repeated measures ANOVA would be more appropriate than t-tests for within-subjects analyses.

3) The authors should explain, possibly in the methods, why NE/DA and Epi/DA are measures of sympathetic tone. This ratio is not a measurement I commonly come across in stress research.

4) Table 3 is a little difficult to follow. It's not clear how baseline measurements were obtained or if "baseline" is the right word to describe this condition, as baseline implies to me a within-subjects approach was used, which I do not think is the case.

5) The “Biochemical measurements” section nearly completely lacks methodological detail or a citation containing these details for each assay.

6) Overall, while this manuscript is easy to follow for the most part, it could use some close editing for conciseness and colloquial language throughout. This is most noticeable in the discussion, which is not currently written in a manner ready for publication. The discussion needs close editing for content and language. The discussion alternates between several long paragraphs that can go on for pages and short paragraphs containing 2-3 sentences (e.g. line 275). The long paragraphs lack focus by presenting multiple arguments, each of which could be broken into its own paragraph and discussed in detail. The short paragraphs lack the context to understand why the findings discussed are important additions to scientific literature. Overall, the authors need to derive a clear story from their data and highlight what is novel and important additions to the body of literature on stress physiology and nutrition. Moreover, this should be completed while considering the possible confounds of changes in physical activity levels due to repeated cage changes.

Reviewer #3: In this manuscript, “Biological responses including changes of amino acids in plasma in mice subjected to repeated cage exchange stress”, Fushuku and colleagues use an acute model of repeated cage exchange model to examine the impact of acute psychological stress on urinary stress and plasma metabolic measures (in addition to various physiological measures). This is an interesting paper with some interesting data; however, I have several concerns with the paper.

These include:

- Lack of details within the introduction relating to the psychological stress-nutrient link, how this this may influence dietary outcomes (links to obesity etc.), and why it is important to study this.

- Lack of details on the model and statistical analysis techniques used for each parameter. Like figure legends, all table legends should also include what analysis was completed on the data presented in the corresponding table. It would also be appropriate to include all statistical analysis data, including F and p values completed for all data in a supplementary file.

- Lack of information in the discussion to agree/disagree with observed results. At this point in time, the discussion is mainly composed of result repetition with limited arguments that can be made to agree/disagree with the observations in the current study. I can appreciate that there may be limited evidence to support some of the observed results, however there are many sections where evidence is published and should be discussed.

- I also have concerns with grammar and how this paper reads at present– I would like to request that the authors obtain professional editing of this article prior to revision.

Please see comments below for more details:

Comment 1:

Title: The focus of the paper is about nutrient-related and metabolic changes in response to acute stress. In addition, this study was only completed in male mice and as such, the title should reflect this. I propose that the title should read “Acute Repeated Cage Exchange Stress Modifies Urinary Stress and Plasma Metabolic Profiles in Male Mice”

Comment 2:

Lines 66-69. This focus of this study is investigating the link between psychological stress and nutrient profiles, yet the introduction only provides very limited overview of this. Please add more information into the introduction regarding this. Some articles that may be of interest can be found below:

Sarris J, Logan AC, Akbaraly TN, Amminger GP, Balanzá-Martínez V, Freeman MP, Hibbeln J, Matsuoka Y, Mischoulon D, Mizoue T, Nanri A, Nishi D, Ramsey D, Rucklidge JJ, Sanchez-Villegas A, Scholey A, Su KP, Jacka FN; International Society for Nutritional Psychiatry Research. Nutritional medicine as mainstream in psychiatry. Lancet Psychiatry. 2015 Mar;2(3):271-4. doi: 10.1016/S2215-0366(14)00051-0.

Lopresti AL. The Effects of Psychological and Environmental Stress on Micronutrient Concentrations in the Body: A Review of the Evidence. Adv Nutr. 2020 Jan 1;11(1):103-112. doi: 10.1093/advances/nmz082.

Please also add in description about the importance of studying impact of psychological stress on nutrient and metabolic profiles and why understanding the mechanisms behind these changes is important in improving human health e.g., potential links to metabolic diseases such as diabetes etc.

Comment 3:

Lines 70-75: The description of the variety of rodent models that have been created/used in studying the psychological stress-nutrient profiles is limited in this article. Please add in more information and details about other studies. For example, the first line of this paragraph (Line 70) notes ‘psychological and physical stress’ but only proceeds to describe physical stressors. Please provide additional details for psychological stressors that are used in rodents and their impact on nutrient profiles.

Comment 4:

Line 74: What is the importance of a decreased amino acid profile in response to stress, please elaborate on this point. It would be appropriate to give a brief description of amino acid metabolism here. In addition, it would be ideal to discuss previous publications that have described how amino acid metabolism is impacted by psychological stress, for e.g.., Milakofsky L, Harris N, Vogel WH. Effect of repeated stress on a number of plasma amino acids and related compounds in young and old rats. Physiol Behav. 1996 Sep;60(3):969-71. doi: 10.1016/0031-9384(96)00094-7.

Comment 5:

Line 76: What is the importance of increased body temperature in response to stress, please elaborate on this point.

Comment 6:

Line 85: It is unclear why this paper focuses on amino acids. The importance of stress-related impacts on amino acids needs to be further highlighted within the introduction.

Comment 7:

Line 102: I assume ‘25-30g’ is the weight of the mice, please ensure this is moved into a more appropriate section within the methods. It might be appropriate to write a brief description (only needs one sentence) stating that mice weights ranged between 25-30g at the start of experiment.

Comment 8:

Line 114: Please add in centrifugation settings including speed, temperature, and time used

Comment 9:

Line 118: Please clarify what is meant by ‘novel mice’? All mice from the CT and CE groups should be included in all tables and figures. Please ensure naming is kept consistent for control and stressed animals throughout text.

Comment 10:

Line 121-127: You need to clarify that this stress was only completed once and is an acute stress paradigm

Comment 11:

Line 125-127: This should be re-worded. For e.g To ensure consistent food and water conditions across all groups, control mice were fasted for 8h concomitantly with the stressed mice.

Comment 12:

Line 131: Please add in that the water measurement protocol has already been published. For e.g. add in “……….. based on previously published methods [16]”

Comment 13:

Line 136: Briefly describe what the elevated plus maze tests for? What do the open and closed arms represent (unsafe versus safe environments) and what does spending more time in the open arms mean?

Line 141: How was the elevated plus maze recorded? It needs to be clarified if this test was recorded by a camera or observed by a person throughout the duration of the test. The reference used in the text (Anchan et al) digitally records this test. If the current study used digital recording, please add in appropriate details of camera used. If no camera was used to record the test, how did you minimize the risk of disturbing the outcome of the test with a person in the room? Please also add in information about how the behavioural data was analysed prior to statistical analysis, which would either be via automated behavioural program or manually processed. If the data was manually scored, please include details as to how bias was controlled (blinded analysis, etc…) There is also no mention of the light levels used in the room for the behavioural test. Many studies have noted of the significant impact that light levels can have on behavioural testing, for e.g. Peirson SN, Brown LA, Pothecary CA, Benson LA, Fisk AS. Light and the laboratory mouse. J Neurosci Methods. 2018 Apr 15;300:26-36. doi: 10.1016/j.jneumeth.2017.04.007. Please add the light level conditions of the behavioural room into this section.

You also need to include justification for the use of only the one behavioural test, elevated plus maze. It would also be appropriate to include that having only one behavioural test is a limitation, and as such needs to be added into the limitations section. Additionally, the data from the EPM can only provide information for “anxiety-like behaviour” rather than anxiety. Please re-word the paper to ensure that you are describing the behaviour as “anxiety-like” rather than “anxiety”.

Comment 14:

Line 146-148: Please include a brief description of the volume loading range (µL of plasma loaded) for plasma samples used in the commercial kits and HPLC.

Comment 15:

Line 177-178: I am unclear what is meant by this sentence “the temperature at 120 min was also evaluated as that at 0 min for the next CE”. Please clarify and re-word if necessary.

Comment 16:

Line 182: Is this meant to read “higher at 08:00h than at 16:00h” rather than “higher at 08:00h than at 08:00h”? Please check this sentence and change accordingly.

Comment 17:

Line 190: You need to discuss why these parameters are related to psychological stress. Alternatively, remove this sentence from the results and move to the discussion section where you should elaborate on why the change you observed to physiological parameters are related to stress. Please include appropriate references when discussing.

Comment 18:

Line 266-267: As you have discussed changes to BT earlier in this paragraph, it would be appropriate to discuss the impact that altered SNS analytes (and therefore a presumably altered SNS state) has on BT. There is a lot of literature describing these links. Here is a paper that may be of interest, Tan CL, Knight ZA. Regulation of Body Temperature by the Nervous System. Neuron. 2018 Apr 4;98(1):31-48. doi: 10.1016/j.neuron.2018.02.022

Comment 19:

Lines 280-281: You need to state which four amino acids were increased with CE.

Comment 20:

Line 315-316: How are amino acids impacted by reduced glucose levels? What are the mechanistic links between glucose and amino acids? Please add in more information.

Comment 21:

Table 1: There is an error in the table title, I assume the title should read “body weight changes in the CT and CE mice”. In addition, analysis for this data should consist of two-way repeated measures ANOVA, followed by post hoc comparisons if the main effect of STRESS or TIME or interaction (stress*time) effect is found to be significant. Please update analysis and include details of statistical analysis into the table legend.

Comment 22:

Table 3: I have some confusion around the “baseline” column. Is this baseline values for the CT or CE mice? If these are baseline measures for matched animals, then these values need to be separated as either baseline for CT or baseline for CE. In addition, a paired two-way repeated measures ANOVA needs to be completed for these parameters.

Alternatively, is this another group of animals that were used, perhaps a control group that weren’t food or water restricted? If so, then a one-way ANOVA needs to be completed. Please clarify this in the table and in the methods sections and rename this group as the term “baseline” has been used throughout the text for different reasons. If this is another group of animals, why are the “baseline” animals not included in other measures such as psychological (behavioural) or body weight parameters? Also, please stipulate the methods for weighing organs in the methods section.

Comment 23:

Figure 1: In addition to examining the time effect, it would be appropriate (and interesting) to examine the stress effect (and interaction effect) for BT of the CT and CE groups. Please update analysis, which would include two way repeated measured ANOVA on this data and update discussion accordingly. The same comment goes for urinary catecholamine and creatine measures (Figure 2), please update statistical analysis, figures and discussion accordingly.

Comment 23:

In terms of all data, why do the n values change from n=5 to n=10 across the various figures and tables. You need to include justification for why animals were removed from certain analyses, which might include outlier testing, etc. Please include information about this in the methods section.

6. PLOS authors have the option to publish the peer review history of their article (what does this mean?). If published, this will include your full peer review and any attached files.

Reviewer #1: No

Reviewer #2: No

Reviewer #3: **Yes: **Tessa Helman

---

## [Author Response · Author response to Decision Letter 0]

11 Aug 2023

Yukiori Goto, Ph.D.

Academic Editor

PLOS ONE 

Dear Prof. Goto,

We are pleased to learn that our manuscript (PONE-D-23-13098, Acute repeated cage exchange stress modifies urinary stress and plasma metabolic profiles in male mice) can be considered for publication in your journal.

Journal Requirements

We have followed PLOS ONE’s requirements for the manuscript. In particular, we followed the font size requirements in the respective locations and the file naming requirements.

"This work was supported by grants from the Ministry of Education, Culture, Sports, Science, and Technology of Japan (#20K10430 to Miharu Ushikai) and funded by the Kodama Memorial Fund for Medical Science Research. This work was also supported by the Core Research Program for “Preventive Medicine Core Unit” by Kagoshima University (to Masahisa Horiuchi)." Please state what role the funders took in the study. If the funders had no role, please state: "The funders had no role in study design, data collection and analysis, decision to publish, or preparation of the manuscript."

We added the following sentences to the Acknowledgments (lines 417–425 of the revised version without corrections): "This work was supported by grants from the Ministry of Education, Culture, Sports, Science, and Technology of Japan (#20K10430 to Miharu Ushikai) and funded by the Kodama Memorial Fund for Medical Science Research. This work was also supported by the Core Research Program for “Preventive Medicine Core Unit” by Kagoshima University (to Masahisa Horiuchi). The funders had no role in study design, data collection and analysis, decision to publish, or preparation of the manuscript." was also added to the section.

We have uploaded our study’s minimal underlying data set as Supporting Information Files.

We added the ethics statement to the “Materials and Methods” section (lines 143–145).

This study was approved by the Ethics Committee for Animal Experimentation at Kagoshima University (approval numbers MD20122, and MD21066), in accordance with Japanese national guidelines for animal experiments. 

Reviewer #1: This study suggests that repeated cage exchange induce physiological and biochemical alterations. Although the results sound very interesting, there are several concerns below that would help to fully appreciate the study.

Major concerns:

1. Water intakes could be one of crucial indicators reflecting physiological alterations induced by repeated cage exchange (CE). This issue has been described on line 186-187, but without data, even though it is also mentioned that water intakes were measured every 2 hours on line 131-133. 

We showed the data after analysis using two-way repeated measures of ANOVA as follows. 

CE mice, n = 5, 0–2 h: 0.038 ± 0.015 g, 2–4 h: 0.004 ± 0.004 g, 4–6 h: 0.032 ± 0.010 g, 6–8 h: 0.046 ± 0.010 g; CT mice, n = 5, 0–2 h: 0.000 ± 0.000 g, 2–4 h: 0.006 ± 0.004 g, 4–6 h: 0.000 ± 0.000 g, 6–8 h: 0.016 ± 0.008 g, F(3,24) = 2.23, p = 0.111. The p value did not reach statistical significance possibly because of the variation of the data. Therefore, we described the comparison based on the data for 0–8 h water intake in CE and CT mice using an unpaired t-test because normality and equal contribution were maintained. The number of mice was corrected.

2. NE/DA and Epi/DA ratios were evaluated based on the reason that cage exchange influenced on Cr concentration (line 199-203), although other factors such as corticosterone was still evaluated relative to Cr. Since cage exchange influenced on Cr, it should affect all factors uniformly, such that it does not much make sense that NE and Epi are evaluated relative to DA but not others.

This point was also raised by the reviewer 2. We deleted the data regarding evaluation relative to DA. The related sentences were changed because of this deletion. 

3. This study has reported that decreased liver and brown adipose tissues in both CE and CT models only in 8 hours, which is surprisingly fast metabolism beyond the ordinary understanding. Moreover, in the CE model, blood glucose level, but not glucagon and insulin, was decreased. Therefore, decreased liver and brown adipose tissues could not be explain only by glycogen contents. Providing details to explain this observation would be helpful to understand the study. Taking a ratio of tissue to whole body weights would also yield an additional insight on this issue.

We added the data regarding the tissue to whole body weight ratio to Table 3. Both groups still showed decreases in the ratios of liver and BAT weight to body weight compared with the value at 08:00 h. There was no significant difference in insulin levels, but the insulin level was lower at 16:00 h than that at 08:00 h in both groups of mice. At present, we speculate that a decrease in glycogen is the main cause of the decreased weights of the liver and BAT at 16:00 h compared with that at 08:00 h.

Therefore, we added the following sentences to the Discussion (lines 333–336).

No significant difference in insulin was observed, but insulin levels were lower at 16:00 h than at 08:00 h in both groups of mice. At present, we speculate that the decrease in glycogen is the main cause of the lower weights of the liver and BAT at 16:00 h than at 08:00 h. 

Reviewer #2: The following study used repeated cage exchange in mice as a model of psychological stress to investigate the outcomes of stress exposure on body temp., organ weight, anxiety behaviors, and amino acid concentrations in urine. The manuscript is written clearly enough to follow and the experimental design seems appropriate, although there are some concerns about the statistical methods that require attention. My biggest concern with this study is the use of the cage exchange model of stress, as it contains many confounds that cannot be easily controlled for. This manuscript could be made suitable for publication, but would require an overhaul of the discussion and a very careful consideration of the rationale for the experimental treatments.

1) My major concern with this study is that the authors attribute their dependent outcomes to “psychological stress” due to repeated cage changes. This interpretation is problematic. Mice typically become more active in new environments, such as following cage changes. Can the authors tease apart the effects of stress from those of physical activity? Nearly every single dependent measure (e.g. water drinking, body temperature, changes to amino acids, cort levels, etc.) could just as easily be attributed to increased physical activity. This is an issue that I’m not sure can be readily addressed. Minimally, the authors may be required to restructure their arguments in the discussion to reflect this possible confound or provide evidence from literature why this might not be an issue for particular dependent measures (like with amino acid levels, if even possible). 

We agree with this comment. 

The measures (e.g., water drinking, body temperature, changes in amino acids, corticosterone levels) could easily be attributed to increased physical activity. At present, we cannot distinguish the effects of stress from those of physical activity. Therefore, we added the following sentences to the Abstract and Discussion sections (line 21 and lines 280–286).

In the Abstract: “Exposure to a novel environment is psychologically and physically stressful for humans and animals.” 

In the Discussion: “However, these measures (water drinking, BT, changes in amino acids) could also be attributed to increased physical activity. At present, we cannot differentiate the effects of stress from those of physical activity. The direct relationship between the changes in amino acids and psychological findings, including urinary stress and anxiety-like test results, should be evaluated by interference experiments.”

Second, the use of “psychological” to describe behavioral testing and outcomes, as well as “psychological stress” is further hard to tease apart using a rodent model. How do the authors know their outcomes are purely psychological?

We agree with this comment. As the reviewer noted, we could not state that these outcomes are purely psychological. Therefore, we added the following sentences to the Abstract and Discussion section (line 21 and lines 280–286) as well as in response to the previous query.

In the Abstract: “Exposure to a novel environment is psychologically and physically stressful for humans and animals.” 

In the Discussion: “However, these measures (water drinking, BT, changes in amino acids) could also be attributed to increased physical activity. At present, we cannot differentiate the effects of stress from those of physical activity. The direct relationship between the changes in amino acids and psychological findings, including urinary stress and anxiety-like test results, should be evaluated by interference experiments.”

2) Statistical considerations: 

① Test statistics are nearly completely absent from the results section. The authors should report their test statistics and make it clear which tests were being performed since the authors chose to perform parametric and non-parametric tests. 

② How did the authors determine the normality and non-normality of data? Most, if not all, of the measurements are continuous variables that should fall within a normal distribution. Thus, it is unclear how the assumption of normality is violated. 

③ If I am following the statistical analysis section correctly, it seems that a repeated measures ANOVA would be more appropriate than t-tests for within-subjects analyses.

We submitted all raw data as supplementary information. In addition, we submitted the results regarding normality and homogeneity of variance as supporting information files. We determined the normality of data with the Shapiro-Wilk normality test using EZR software. The data in Fig 1 and Table 1 were statistically analyzed using two-way repeated measures ANOVA. We added the results to the respective legends. For Fig 2, some repeated measurements were missing, namely urine samples. Therefore, we did not perform two-way repeated measures ANOVA.

3) The authors should explain, possibly in the methods, why NE/DA and Epi/DA are measures of sympathetic tone. This ratio is not a measurement I commonly come across in stress research.

This point was also raised by reviewer 1. We deleted the data regarding evaluation relative to DA. Therefore, we changed these sentences. 

4) Table 3 is a little difficult to follow. It's not clear how baseline measurements were obtained or if "baseline" is the right word to describe this condition, as baseline implies to me a within-subjects approach was used, which I do not think is the case.

We replaced “baseline” with “08:00 h” to avoid confusion.

5) The “Biochemical measurements” section nearly completely lacks methodological detail or a citation containing these details for each assay.

We added the explanations to the Materials and Methods section.

6) Overall, while this manuscript is easy to follow for the most part, it could use some close editing for conciseness and colloquial language throughout. This is most noticeable in the discussion, which is not currently written in a manner ready for publication. The discussion needs close editing for content and language. The discussion alternates between several long paragraphs that can go on for pages and short paragraphs containing 2-3 sentences (e.g. line 275). The long paragraphs lack focus by presenting multiple arguments, each of which could be broken into its own paragraph and discussed in detail. The short paragraphs lack the context to understand why the findings discussed are important additions to scientific literature. Overall, the authors need to derive a clear story from their data and highlight what is novel and important additions to the body of literature on stress physiology and nutrition. Moreover, this should be completed while considering the possible confounds of changes in physical activity levels due to repeated cage changes.

We thank you for these important comments. 

According to the comments, we added the following sentences to the Discussion section (lines 352–356). In addition, we separated the long paragraph into several parts. 

The outcomes, such as increased BT induced by repeated CE, might be related to increased physical activity. However, in the literature, exercise inducing physical activity reduced plasma BCAA levels in diabetic mice [19] and normalized plasma BCAAs in healthy overweight humans [37]. At present, we suggest that the increased BCAA concentration in CE mice is not explained by increased physical activity.

Reviewer #3: In this manuscript, “Biological responses including changes of amino acids in plasma in mice subjected to repeated cage exchange stress”, Fushuku and colleagues use an acute model of repeated cage exchange model to examine the impact of acute psychological stress on urinary stress and plasma metabolic measures (in addition to various physiological measures). This is an interesting paper with some interesting data; however, I have several concerns with the paper.

These include:

- Lack of details within the introduction relating to the psychological stress-nutrient link, how this this may influence dietary outcomes (links to obesity etc.), and why it is important to study this.

We agree that the importance of the psychological stress-nutrient link should be described. 

We added the following sentences to the Introduction (lines 92–95).

If a link exists between psychological stress and nutrients, it may be possible to use nutrients to manage psychologically stressful conditions. To achieve this goal, the changes in nutrients that occur under psychological stress should be determined.

- Lack of details on the model and statistical analysis techniques used for each parameter. Like figure legends, all table legends should also include what analysis was completed on the data presented in the corresponding table. It would also be appropriate to include all statistical analysis data, including F and p values completed for all data in a supplementary file.

We performed two-way repeated measures ANOVA for Fig1 and Table 1. We described the information related to the statistical procedure in the legends of the figures and tables. We submitted all statistical analysis data, including the F and p values calculated for all data as supporting information.

- Lack of information in the discussion to agree/disagree with observed results. At this point in time, the discussion is mainly composed of result repetition with limited arguments that can be made to agree/disagree with the observations in the current study. I can appreciate that there may be limited evidence to support some of the observed results, however there are many sections where evidence is published and should be discussed.

As the reviewer noted, we should avoid repetition of results with limited arguments. Based on newly added references, we included a more thorough discussion of the results observed in the present study than that included in the previous version.

We added the following sentences to the Discussion (lines 323–331).

Animal stress models subjected to restraint or blood collection showed increased blood glucose levels [33, 34]. Although the blood glucose level is physiologically maintained by hepatic glycogenolysis and gluconeogenesis, those systems are enhanced under stress conditions [33, 34]. However, the present study showed inconsistent results, revealing relatively low blood glucose levels during 8 h of fasting (Table 3). In particular, gluconeogenesis might not be enhanced in the present stress model. Interleukin (IL)-6 plays an important role in enhanced gluconeogenesis in model animals with stress induced by restraint or blood collection, and further studies will be required to examine IL-6 in the repeated CE stress model.

- I also have concerns with grammar and how this paper reads at present– I would like to request that the authors obtain professional editing of this article prior to revision.

The manuscript underwent professional editing to improve the grammar and enhance clarity.

Comment 1:

Title: The focus of the paper is about nutrient-related and metabolic changes in response to acute stress. In addition, this study was only completed in male mice and as such, the title should reflect this. I propose that the title should read “Acute Repeated Cage Exchange Stress Modifies Urinary Stress and Plasma Metabolic Profiles in Male Mice”

We agree with this comment. The title was changed as suggested.

Comment 2:

Lines 66-69. This focus of this study is investigating the link between psychological stress and nutrient profiles, yet the introduction only provides very limited overview of this. Please add more information into the introduction regarding this. Some articles that may be of interest can be found below:

Sarris J, Logan AC, Akbaraly TN, Amminger GP, Balanzá-Martínez V, Freeman MP, Hibbeln J, Matsuoka Y, Mischoulon D, Mizoue T, Nanri A, Nishi D, Ramsey D, Rucklidge JJ, Sanchez-Villegas A, Scholey A, Su KP, Jacka FN; International Society for Nutritional Psychiatry Research. Nutritional medicine as mainstream in psychiatry. Lancet Psychiatry. 2015 Mar;2(3):271-4. doi: 10.1016/S2215-0366(14)00051-0.

Lopresti AL. The Effects of Psychological and Environmental Stress on Micronutrient Concentrations in the Body: A Review of the Evidence. Adv Nutr. 2020 Jan 1;11(1):103-112. doi: 10.1093/advances/nmz082.

We agree with the comments. However, the study by Sarris et al. was previously cited as reference number 6. We added the study by Lopresti as reference 7. 

We added the following sentence to the Introduction (lines 64–66). 

Limited information on the relationship between resilience to stress and nutritional factors, including amino acids, has been reported, although amino acids-related compounds and minerals are thought to confer stress resilience [6, 7]. 

Please also add in description about the importance of studying impact of psychological stress on nutrient and metabolic profiles and why understanding the mechanisms behind these changes is important in improving human health e.g., potential links to metabolic diseases such as diabetes etc.

We added the following sentences to the Introduction (lines 92–97). 

If a link exists between psychological stress and nutrients, it may be possible to use nutrients to manage psychologically stressful conditions. To achieve this goal, the changes in nutrients that occur under psychological stress should be determined. Moreover, psychological conditions are potentially linked to the prognosis of metabolic diseases such as diabetes. Relief of psychological conditions by nutritional intervention might be important in managing metabolic diseases.

Comment 3:

Lines 70-75: The description of the variety of rodent models that have been created/used in studying the psychological stress-nutrient profiles is limited in this article. Please add in more information and details about other studies. For example, the first line of this paragraph (Line 70) notes ‘psychological and physical stress’ but only proceeds to describe physical stressors. Please provide additional details for psychological stressors that are used in rodents and their impact on nutrient profiles.

We added the following sentences to the Introduction to provide additional details regarding the nutrient profiles for psychological stressors using rodent models based on the new reference (lines 73–78). 

These procedures include isolating mice or housing two mice together to generate dominant and subordinate mice. Regarding energy expenditure, dominant mice have a significantly greater energy expenditure than subordinate mice at the end of chronic psychosocial stress procedures. Additionally, subordinate mice show higher plasma glucose and non-esterified fatty acid levels than dominant mice after stress procedures. However, information on amino acid profiles is limited.

Comment 4:

Line 74: What is the importance of a decreased amino acid profile in response to stress, please elaborate on this point. It would be appropriate to give a brief description of amino acid metabolism here. In addition, it would be ideal to discuss previous publications that have described how amino acid metabolism is impacted by psychological stress, for e.g.., Milakofsky L, Harris N, Vogel WH. Effect of repeated stress on a number of plasma amino acids and related compounds in young and old rats. Physiol Behav. 1996 Sep;60(3):969-71. doi: 10.1016/0031-9384(96)00094-7.

We added the following sentence to the Discussion section regarding the reference suggested by the reviewer (lines 78–81).

In addition, in one study using an immobilization model where mice were restrained in a tube for several hours, the levels of almost all amino acids in plasma were reported to be decreased [10], and several amino acids were altered depending on the age of mice and duration of stress [11].

Comment 5:

Line 76: What is the importance of increased body temperature in response to stress, please elaborate on this point.

We added the following sentence (lines 83–84). 

The importance of increased BT in response to stress may be that it alters brain function [13].

Comment 6:

Line 85: It is unclear why this paper focuses on amino acids. The importance of stress-related impacts on amino acids needs to be further highlighted within the introduction.

We added the following sentence (lines 98–102).

In this paper, we report the changes of nutrients, especially amino acids, which have limited information regarding their responses to stress and are consumed to meet energy demand, in plasma upon establishing a stress model in which the animals can still move freely, namely, repeated cage exchange (CE), mimicking exposure to a novel environment. 

Comment 7:

Line 102: I assume ‘25-30g’ is the weight of the mice, please ensure this is moved into a more appropriate section within the methods. It might be appropriate to write a brief description (only needs one sentence) stating that mice weights ranged between 25-30g at the start of experiment.

This was described (lines 121–122).

Mice that weighed between 25 and 32 g at the start of the experiment were used.

Comment 8:

Line 114: Please add in centrifugation settings including speed, temperature, and time used

We added this information (3000 rpm for 10 min at room temperature) (line 134).

Comment 9:

Line 118: Please clarify what is meant by ‘novel mice’? All mice from the CT and CE groups should be included in all tables and figures. Please ensure naming is kept consistent for control and stressed animals throughout text.

We changed the word from “novel” to “different” to reduce confusion.

We described how mice were used in detail (lines 138–143).

A total of 72 mice were used to obtain the results for Fig 1 (10 control (CT) and 10 CE mice), Fig 2 (10 different CT and 10 different CE mice), Table 1, Table 3 (six different mice were used to obtain data at 08:00 h, six CT and six CE mice from the experiment for Fig 2 were used 1–2 weeks after the experiment), and Table 2 (eight different CT and eight different CE mice). For the water intake experiment, 5 CT and 5 CE mice were used. 

Comment 10:

Line 121-127: You need to clarify that this stress was only completed once and is an acute stress paradigm

We added a sentence to clarify the stress condition (line 148).

This stress was only performed once a day and is an acute stress paradigm.

Comment 11:

Line 125-127: This should be re-worded. For e.g To ensure consistent food and water conditions across all groups, control mice were fasted for 8h concomitantly with the stressed mice.

We replaced the sentences with the wording suggested by the reviewer. 

Comment 12:

Line 131: Please add in that the water measurement protocol has already been published. For e.g. add in “……….. based on previously published methods [16]”

We added the suggested phrase to the sentence.

Comment 13:

Line 136: Briefly describe what the elevated plus maze tests for? What do the open and closed arms represent (unsafe versus safe environments) and what does spending more time in the open arms mean?

We added sentences describing the meaning of the elevated plus maze test (lines 162–165). 

The elevated plus maze test was performed to evaluate anxiety-like conditions. The open and closed arms represent unsafe and safe environments, respectively. Spending more time in the open arms indicates mice are more resistant to anxiety-like conditions [18].

Line 141: How was the elevated plus maze recorded? It needs to be clarified if this test was recorded by a camera or observed by a person throughout the duration of the test. The reference used in the text (Anchan et al) digitally records this test. If the current study used digital recording, please add in appropriate details of camera used. If no camera was used to record the test, how did you minimize the risk of disturbing the outcome of the test with a person in the room? Please also add in information about how the behavioural data was analysed prior to statistical analysis, which would either be via automated behavioural program or manually processed. If the data was manually scored, please include details as to how bias was controlled (blinded analysis, etc…) There is also no mention of the light levels used in the room for the behavioural test. Many studies have noted of the significant impact that light levels can have on behavioural testing, for e.g. Peirson SN, Brown LA, Pothecary CA, Benson LA, Fisk AS. Light and the laboratory mouse. J Neurosci Methods. 2018 Apr 15;300:26-36. doi: 10.1016/j.jneumeth.2017.04.007. Please add the light level conditions of the behavioural room into this section.

We added sentences describing the elevated plus maze experiment and data recording (lines 173–177).

The procedure was performed under a light condition of 15 lx measured by a digital illuminometer (LX-100, Satotech, Kanagawa, Japan). The study was recorded on digital media using a specialized camera (Logicool C615n HD　webcam, Logicool, Tokyo, Japan). Then, the recorded data were analyzed using ANYmaze software (Stoelting, Wood Dale, IL, USA). 

You also need to include justification for the use of only the one behavioural test, elevated plus maze. It would also be appropriate to include that having only one behavioural test is a limitation, and as such needs to be added into the limitations section. Additionally, the data from the EPM can only provide information for “anxiety-like behaviour” rather than anxiety. Please re-word the paper to ensure that you are describing the behaviour as “anxiety-like” rather than “anxiety”.

We added a description to the limitations (lines 399–401). 

Third, we performed only one behavioral test, the elevated plus maze. To evaluate anxiety or anxiety-like behavior precisely, another test should be added in the future. 

Comment 14:

Line 146-148: Please include a brief description of the volume loading range (µL of plasma loaded) for plasma samples used in the commercial kits and HPLC.

We added this information.

Comment 15:

Line 177-178: I am unclear what is meant by this sentence “the temperature at 120 min was also evaluated as that at 0 min for the next CE”. Please clarify and re-word if necessary.

We apologize for the confusing wording in this sentence. We deleted the following sentence. 

The temperature at 120 min was also evaluated as that at 0 min for the next CE.

Comment 16:

Line 182: Is this meant to read “higher at 08:00h than at 16:00h” rather than “higher at 08:00h than at 08:00h”? Please check this sentence and change accordingly.

We corrected the sentence. 

In the case of the BT without CE, the values were significantly higher after 08:10 h than at 08:00 h, possibly because of food removal.

Comment 17:

Line 190: You need to discuss why these parameters are related to psychological stress. Alternatively, remove this sentence from the results and move to the discussion section where you should elaborate on why the change you observed to physiological parameters are related to stress. Please include appropriate references when discussing.

We moved the sentence to the Discussion. As the reviewer noted, increased corticosterone in urine is related not only to psychological stress, but also to physical stress such as exercise (lines 319–320).

Additionally, CE mice had a significantly higher urinary corticosterone level than CT mice (see Results). 

Comment 18:

Line 266-267: As you have discussed changes to BT earlier in this paragraph, it would be appropriate to discuss the impact that altered SNS analytes (and therefore a presumably altered SNS state) has on BT. There is a lot of literature describing these links. Here is a paper that may be of interest, Tan CL, Knight ZA. Regulation of Body Temperature by the Nervous System. Neuron. 2018 Apr 4;98(1):31-48. doi: 10.1016/j.neuron.2018.02.022

We appreciate the suggestion of this reference.

We added the following sentence (lines 312–313).

Enhancement of sympathetic nervous system activity may increase BT through BAT [13]. 

Comment 19:

Lines 280-281: You need to state which four amino acids were increased with CE.

 We added the information to the text. 

Comment 20:

Line 315-316: How are amino acids impacted by reduced glucose levels? What are the mechanistic links between glucose and amino acids? Please add in more information.

We agree with the comment. We added the following sentences describing the role of amino acids in maintaining the glucose level (lines 369–372). 

BCAAs and Phe are used as energy sources [39]. Among the four amino acids that were increased in CE mice, Val, Ile, and Phe are classified as gluconeogenic amino acids that are involved in maintaining the blood glucose level through gluconeogenesis under fasting conditions.

Comment 21:

Table 1: There is an error in the table title, I assume the title should read “body weight changes in the CT and CE mice”. In addition, analysis for this data should consist of two-way repeated measures ANOVA, followed by post hoc comparisons if the main effect of STRESS or TIME or interaction (stress*time) effect is found to be significant. Please update analysis and include details of statistical analysis into the table legend.

We changed the title of Table 1. In addition, we performed two-way repeated measures ANOVA, which showed a significant interaction between the body weights of CT and CE mice during the time course.

Comment 22:

Table 3: I have some confusion around the “baseline” column. Is this baseline values for the CT or CE mice? If these are baseline measures for matched animals, then these values need to be separated as either baseline for CT or baseline for CE. In addition, a paired two-way repeated measures ANOVA needs to be completed for these parameters.

Alternatively, is this another group of animals that were used, perhaps a control group that weren’t food or water restricted? If so, then a one-way ANOVA needs to be completed. Please clarify this in the table and in the methods sections and rename this group as the term “baseline” has been used throughout the text for different reasons. If this is another group of animals, why are the “baseline” animals not included in other measures such as psychological (behavioural) or body weight parameters? Also, please stipulate the methods for weighing organs in the methods section.

This point was also raised by reviewer 2. 

We changed the word “baseline” to “08:00 h”. 

Comment 23:

Figure 1: In addition to examining the time effect, it would be appropriate (and interesting) to examine the stress effect (and interaction effect) for BT of the CT and CE groups. Please update analysis, which would include two way repeated measured ANOVA on this data and update discussion accordingly. The same comment goes for urinary catecholamine and creatine measures (Figure 2), please update statistical analysis, figures and discussion accordingly.

We performed two-way repeated measures ANOVA for the data in Fig 1. There was a significant value for the interaction of stress x time. For Fig 2, some repeated measurements were missing, namely urine samples. Therefore, we did not perform two-way repeated measures ANOVA.

Comment 24:

In terms of all data, why do the n values change from n=5 to n=10 across the various figures and tables. You need to include justification for why animals were removed from certain analyses, which might include outlier testing, etc. Please include information about this in the methods section.

We performed the experiments with different groups of mice that we purchased from the supplier. An explanation was added to the Materials and Methods section (lines 138–143).

A total of 72 mice were used to obtain the results for Fig 1 (10 control (CT) and 10 CE mice), Fig 2 (10 different CT and 10 different CE mice), Table 1, Table 3 (six different mice were used to obtain data at 08:00 h, six CT and six CE mice from the experiment for Fig 2 were used 1–2 weeks after the experiment), and Table 2 (eight different CT and eight different CE mice). For the water intake experiment, 5 CT and 5 CE mice were used.

For confirmation, we submitted most of the raw data as supporting information. In addition, we added the following sentence to the Materials and Methods section (lines 212–214). 

We omitted some data and mice according to outlier tests, including the Smirnov–Grubbs test and QQ plots.

---

## [Decision Letter · Decision Letter 1]

14 Sep 2023

PONE-D-23-13098R1Acute repeated cage exchange stress modifies urinary stress and plasma metabolic profiles in male micePLOS ONE

Dear Dr. Horiuchi,

Thank you for submitting your manuscript to PLOS ONE. After careful consideration, we feel that it has merit but does not fully meet PLOS ONE’s publication criteria as it currently stands. Therefore, we invite you to submit a revised version of the manuscript that addresses the points raised during the review process.

We look forward to receiving your revised manuscript.

Kind regards,

Yukiori Goto, Ph.D.

Academic Editor

PLOS ONE

Journal Requirements:

Additional Editor Comments:

A few minor issues have been raised by the reviewer on the revised manuscript. Please address them, so that I can render acceptance of the manuscript.

Reviewers' comments:

Reviewer's Responses to Questions

**Comments to the Author**

1. If the authors have adequately addressed your comments raised in a previous round of review and you feel that this manuscript is now acceptable for publication, you may indicate that here to bypass the “Comments to the Author” section, enter your conflict of interest statement in the “Confidential to Editor” section, and submit your "Accept" recommendation.

Reviewer #1: All comments have been addressed

Reviewer #3: (No Response)

2. Is the manuscript technically sound, and do the data support the conclusions?

Reviewer #1: Yes

Reviewer #3: Yes

3. Has the statistical analysis been performed appropriately and rigorously? 

Reviewer #1: Yes

Reviewer #3: Yes

4. Have the authors made all data underlying the findings in their manuscript fully available?

Reviewer #1: Yes

Reviewer #3: Yes

5. Is the manuscript presented in an intelligible fashion and written in standard English?

Reviewer #1: Yes

Reviewer #3: Yes

6. Review Comments to the Author

Reviewer #1: The manuscript has been revised according to the comments provided by the reviewers and is considered to be of sufficient quality for publication in this journal.

Reviewer #3: Thank you for your revisions on the manuscript.

A couple of minor notes:

Line 84 – Please add in a sentence to describe a few aspects as to how brain function is altered in response to body temperature changes.

Lines 98-102 – This last section is difficult to read/understand; I would recommend separating into at least two sentences.

Lines 138-142 – It is still difficult to understand which animals were used for each measure. It would be good to insert a diagram here starting from 72 mice, using arrows to separate to different groups of animals that were used for the different experiments/measures. In addition, groups of animals should be represented as n values. For example: Body temperature: Group A includes n=10 CT and n=10 CE (Figure 1); Urinary catecholamines and creatine: Group B includes n=10 CT and n=10 CE (Figure 2); Group C, etc… It would also be appropriate to highlight where a group of animals were used over multiple measures.

Lines 171-172 and 174: This should read ‘anxiety-like behaviour’ rather than ‘anxiety-like conditions’. Please make any other changes to this wording throughout the manuscript.

7. PLOS authors have the option to publish the peer review history of their article (what does this mean?). If published, this will include your full peer review and any attached files.

Reviewer #1: No

Reviewer #3: **Yes: **Tessa Helman

---

## [Author Response · Author response to Decision Letter 1]

22 Sep 2023

Yukiori Goto, Ph.D.

Academic Editor

PLOS ONE 

Dear Prof. Goto,

We are pleased to learn that our manuscript (PONE-D-23-13098, Acute repeated cage exchange stress modifies urinary stress and plasma metabolic profiles in male mice) can be considered for publication in your journal.

Journal Requirements

We have checked the reference list. We have confirmed that it has no papers that have been retracted. We have formatted the list as ordered by the journal.

Additionally, we noticed the error of the version of the EZR for statistical analysis. We changed the number in the revised text.

Reviewer #3: Thank you for your revisions on the manuscript.

A couple of minor notes:

Line 84 – Please add in a sentence to describe a few aspects as to how brain function is altered in response to body temperature changes.

We added the words “for fight-or-flight”.

Lines 98-102 – This last section is difficult to read/understand; I would recommend separating into at least two sentences.

We separated the sentence. 

In this paper, we report the changes in nutrients, especially amino acids, which have limited information regarding their responses to stress and are consumed to meet energy demand. We used a stress model in which the animals can still move freely, namely, repeated cage exchange (CE), mimicking exposure to a novel environment.

Lines 138-142 – It is still difficult to understand which animals were used for each measure. It would be good to insert a diagram here starting from 72 mice, using arrows to separate to different groups of animals that were used for the different experiments/measures. In addition, groups of animals should be represented as n values. For example: Body temperature: Group A includes n=10 CT and n=10 CE (Figure 1); Urinary catecholamines and creatine: Group B includes n=10 CT and n=10 CE (Figure 2); Group C, etc… It would also be appropriate to highlight where a group of animals were used over multiple measures.

We made a diagram showing the way to use mice in the respective experiments. Then, we put it as a supporting figure (S1 Fig), which is indicated in Line 143.

Lines 171-172 and 174: This should read ‘anxiety-like behaviour’ rather than ‘anxiety-like conditions’. Please make any other changes to this wording throughout the manuscript.

We changed the word or the word like that to ‘anxiety-like behavior’ in any places.

---

## [Editor Report · Decision Letter 2]

27 Sep 2023

Acute repeated cage exchange stress modifies urinary stress and plasma metabolic profiles in male mice

PONE-D-23-13098R2

Dear Dr. Horiuchi,

We’re pleased to inform you that your manuscript has been judged scientifically suitable for publication and will be formally accepted for publication once it meets all outstanding technical requirements.

Kind regards,

Yukiori Goto, Ph.D.

Academic Editor

PLOS ONE

---

## [Editor Report · Acceptance letter]

2 Oct 2023

PONE-D-23-13098R2 

Acute repeated cage exchange stress modifies urinary stress and plasma metabolic profiles in male mice 

Dear Dr. Horiuchi:

I'm pleased to inform you that your manuscript has been deemed suitable for publication in PLOS ONE. Congratulations! Your manuscript is now with our production department. 

Kind regards, 

on behalf of

Dr. Yukiori Goto 

Academic Editor

PLOS ONE